# Making geoscientific lab data FAIR: A conceptual model for a geophysical laboratory database

Sven Nordsiek, Matthias Halisch

Leibniz Institute for Applied Geophysics (LIAG), Stilleweg 2, D-30655 Hannover, Germany

*Correspondence to*: Matthias Halisch (matthias.halisch@leibniz-liag.de)

**Abstract.** The term of geoscientific laboratory measurements involves a variety of methods in geosciences. Accordingly, the resulting data comprise many different data types, formats, and sizes, respectively. Handling such a diversity of data, e.g., by storing the data in a generally applicable database, is difficult. Some discipline-specific approaches exist, but a geoscientific laboratory database that is generally applicable to different geoscientific disciplines is missing up to now. However, making

research data available to scientists beyond a particular community has become increasingly important. Global working groups as the Committee on Data of the International Science Council (CODATA) put effort in the development of tools improving research data handling. International standards (e.g., ISO 19156) and ontologies (e.g., UCUM) provide a general framework for certain aspects that are elemental for the development of database models. However, these abstract models need to be adapted to meet the requirements of the geoscientific community. Within a pilot project of the NFDI4Earth initiative, we

developed a conceptual model for a geoscientific laboratory database. For being able to handle complex settings of geoscientific laboratory studies, flexibility and extensibility are key attributes of the presented approach. The model is intended to follow the FAIR data principles to facilitate interdisciplinary applicability. In this study, we consider different procedures from existing database models and include these methods in the conceptual model.

## 1 Introduction

In recent years, transparent and sustainable handling of research data has received increasing attention of different stakeholders, e.g., funding agencies, publishers, and research organisations. Transparency of research data is essential to facilitate reproducibility of scientific results and thus to keep confidence in scientific research (McNutt, 2014). Additionally, available research data can improve the visibility of studies (Piwowar et al., 2007; Colavizza et al., 2020), and enable reuse of once compiled data. Overcoming interdisciplinary obstacles and making research data accessible and usable for scientists from

different disciplines is an important aspect that must be considered, especially in the context of interdisciplinary research projects. Comprehensive programs, e.g., the NFDI initiative that is supported by the German research foundation DFG, have been initiated to promote the development of concepts and infrastructures that help to improve the availability of research data. The growing popularity of machine-learning (ML) algorithms, that mark an innovative way to deal with scientific research data and offer an opportunity to derive indications on hitherto unknown correlations, is another argument to assure availability

of research data. On principal, ML algorithms only perform on large datasets. Therefore, individual laboratory studies that cover only a limited number of samples or field campaigns dealing with a specific case study are not usable in this context. In geoscience, large datasets result from comprehensive projects, e.g., the International Ocean Discovery Program (IODP). Such datasets represent a more suitable object for the application of ML algorithms as they yield a large amount of data measured and processed under uniform conditions. Accordingly, the relevant disciplines from geoscience are quite advanced in sharing

their research data and utilising sophisticated databases (e.g., PANGAEA). However, there are still parts in geoscience where sharing of research data is at the very beginning and appropriate databases are missing up to now, e.g., in geoscientific laboratory research. In this field, approaches for databases exist for single methods (e.g., SIP-Archiv) and for separate disciplines (e.g., Lehnert et al., 2000; Strong et al., 2016; He et al., 2019; Bär et al., 2020). However, up to now, an interdisciplinary geoscientific laboratory database providing research data to a broad scientific community is missing.

For a successful geoscientific laboratory study, different conditions need to be fulfilled in advance. Laboratory measurements require an appropriate (and often expensive) instrumentation as well as laboratory staff trained on the relevant methods. When sample material is rare, obtaining a sufficient amount for the measurements can be an additional problem. Time-consuming, laborious sample preparation and repeated measurements to assure high quality results are other factors that, together with the aforementioned aspects, make geoscientific laboratory data highly valuable. In consideration of these issues, many scientists

have reservations about sharing their own data, as they are afraid of data misuse and an insufficient acknowledgement of their contribution (Tenopir et al., 2018). Although uncertainty about the aspect of intellectual property is an important factor interfering the willingness to share research data, we do not consider this problem, as it would go beyond the scope of our study. We refer to Carroll (2015) and Labastida and Margoni (2020), for instance, where legal aspects of data sharing and licensing of data will be addressed. Instead, we focus here on rather technical issues related with the exchange of data.

According to a survey by Volk et al. (2014), confusion about requested and received data, respectively, is a major problem impeding data sharing between scientists. More precisely, scientists providing data are not sure about which data exactly was requested and those who asked for data have problems to understand the data they received. The FAIR data principles (Wilkinson et al., 2016) can be a solution to this problem and to other issues impeding data sharing. The FAIR data principles work as a guideline to improve sharing of research data, where "FAIR" is an acronym representing the attributes "findable",

"accessible", "interoperable", and "reusable". Each of these attributes is defined in detail by several criteria, so that these principles can be seen as a comprehensive aid to facilitate openness (Bailo et al., 2020; Kinkade and Shepherd, 2022) not only for human access to research data, but also for automatic data collection by machine driven algorithms (Weigel et al., 2020). Despite all the reservations, challenges, and uncertainties as mentioned above, sharing research data is advantageous for both, the scientific community and the individual scientist. Due to the enormous effort required to perform comprehensive and

meaningful geoscientific laboratory experiments, a widespread use of once measured data is highly desirable. Sharing the data with the scientific community is a way to increase the benefit that can be gained from such studies. Making research data accessible allows other scientists to use the existing data to test new models or to apply new approaches for data processing

and evaluation. In the near future, the application of artificial intelligence to large parameter databases may help to discover new relationships in geosciences (Yu and Ma, 2021).

In this study, we present a conceptual database model, particularly designed for geoscientific laboratory data. Nevertheless, this general concept can be adopted towards field scale data with ease. The respective requirements will be discussed in detail in section 2, followed by a short review of existing approaches that deal with distinct aspects of geoscientific database models (section 3). In section 4, we present our conceptual model that is intended to follow the FAIR data principles, as well as recent approaches of modern research data management. With this, access to geoscientific laboratory data will be much more

convenient for scientists from different disciplines in the future.

## 2 Requirements

A model for a geoscientific laboratory database needs to satisfy several requirements resulting from different types of data that will be stored, the variety of targeted users, and the intended field of application of the database. In this section, we describe the requirements in detail.

**2.1 Diversity of data and algorithms**

Geoscientific laboratory investigations comprise many different methods resulting in a variety of data types: Single averaged values, time series, spectral data, and images in 2-D and 3-D, respectively, are examples for typical results of geoscientific laboratory investigations. These different types of data come along with different file formats in which the data are stored, e.g., text files, image files, and other, eventually proprietary, file formats have to be considered. The variety of data types and file

formats induces a wide range of file sizes spreading from few kilobytes, e.g., porosity measurements and spectral induced polarisation (SIP) spectra, to more than 20 gigabytes, e.g., images from micro computed tomography ($\mu$-CT). Reliable handling of the multitude of data types, file formats and sizes is an important challenge within the context of modern research data management, especially for the development of an interdisciplinary applicable geoscientific laboratory database.

For all measured data, distinct software is needed to evaluate the data and to prepare the data prior to evaluation, if necessary.

The processing software can be published under different licenses from open source self-written codes to proprietary programmes. A consequent use of software with open source code is the easiest way to ensure compliance with the FAIR data principles. However, independent of the applicable software licenses, a detailed documentation of the software, its current version and status is mandatory to produce reproducible results.

**2.2 Flexibility and extensibility of the database**

Beside the diversity of laboratory data, the extensibility of the database is another important requirement concerning a laboratory database. In the context of geoscientific applications, extensibility of the database not only refers to the addition of new data. It may also be necessary to incorporate newly developed instruments and methods, modified workflows, additional

samples, and alternative algorithms for the evaluation of measured data, for instance. Therefore, extensibility of the database is imperative to its applicability in daily laboratory routine and must be considered in the first place when developing the database.

International standards on metadata as ISO 19156 and sophisticated ontologies (e.g., Janowicz et al., 2018) enable linking to and exchange with other databases. By following such standards, a database model can be extended to fields beyond the original discipline.

Not only the data related with current studies may need to be extended, but also projects already finished may be reconsidered when new approaches demand a review and methods hitherto unconsidered become relevant. In such a case, existing datasets should be connectable with new data without disarranging the database. Thus, a database intended for the use with geoscientific laboratory studies has to be flexible enough to handle complex and fast growing sets of laboratory data.

## 2.3 Interdisciplinary applicability

Interdisciplinary applicability within geosciences is a key feature of the desired database model. Even when limited to geoscientific disciplines, the scientific language used in geophysics, hydrogeology, and hydrology, for instance, varies and parameters relevant for one discipline may be unfamiliar to researchers from other disciplines. To prevent misinterpretation of the contents of the database and to make it truly interoperable, the discipline-specific differences must be respected while developing the database. The database must be usable independent of the original discipline of the user. Simultaneously, different parameters may characterise similar physical properties on different scales and with different units. In the database, the physical properties must be described straightforward to exclude any misinterpretation. Therefore, a consistent way to express physical properties with suitable parameters and units is needed and transfer of data into the different geoscientific disciplines must be feasible as it is essential for the interoperability of the database.

## 3 An overview on case specific solutions

According to the aforementioned aspects, flexibility, extensibility, and interdisciplinary applicability must be key features of a geoscientific laboratory database. In geoscience and neighbouring fields, some database models exist that take into account at least some of these requirements.

### 3.1 Relational databases

In many geoscientific disciplines, relational databases are used to organise data. For instance, Lehnert et al. (2000) developed a database structure concerning geochemical data. Horsburgh et al. (2008) presented a database model for environmental data. Strong et al. (2016) and He et al. (2019) report on geoanalytical databases. The database presented by Bär et al. (2020) is an excellent approach to store petrophysical data as it comprises a large number of petrophysical properties and provides detailed documentation of the data and data quality with appropriate metadata. The mentioned examples demonstrate that relational

databases provide flexibility due to the modular structure, so that subsequent incorporation of new components is feasible. Although the preceding approaches are well suited for usage in their respective disciplines, the applicability as a general model

for a geoscientific laboratory database is limited. For the incorporation of various geoscientific disciplines, it is important to consider different vocabularies, specific for each discipline. Translation between the individual geoscientific vocabularies is crucial for an interdisciplinary database. Thesauri may be a solution to this problem, as they allow communication across different scientific vocabularies (e.g., Albertoni et al., 2018; Morrill et al., 2021). They further allow a later inclusion of disciplines without changing the framework of the original database.

## 3.2 Complex workflow descriptors

Some laboratory methods are common procedures that follow a certain standard (e.g., DIN, ISO). However, often laboratory measurements are individual experiments with a community-wide accepted workflow, but without any officially defined procedure. These experiments require distinct descriptions of each step from sample preparation to evaluation of the measured data. To guarantee interoperability of research data according to the FAIR data principles, the documentation of workflows

has to be both understandable for researchers from other disciplines and machine-readable. Verdi et al. (2007) exemplarily analyse the procedures related with nuclear magnetic resonance (NMR) spectroscopy and describe a conceptual model to capture the workflow of NMR spectroscopy experiments. Weigel et al. (2020) focus on the findability of data and workflows for machines and emphasize the importance of using persistent identifiers in this context. Samuel and König-Ries (2022) highlight the significance of understandable comprehensive information about the provenance of scientific results and present

an own approach to this information based on the existing standard PROV-O (Lebo et al., 2013).

## 3.3 Homogenization of interdisciplinary physical units

In medical science, the issue of misinterpretation of units has been thoroughly addressed by Schadow et al. (1999). Their solution to this problem is known as "Unified Code for Units and Measure" (UCUM), which is still part of current discussion (Hall and Kuster, 2022). With the UCUM system, each unit of a physical property is described by a vector of seven dimensions.

According to Schadow et al. (1999), these dimensions are length (meters), time (seconds), mass (gram), electrical charge (Coulomb), temperature (Kelvin), luminous intensity (candela), and angle (radians). Similar to SI units, every unit of a physical quantity can be expressed as a combination of these seven basic units.

However, the digital representation of physical units in databases is a problem that is relevant to all fields of science. Hanisch et al. (2022) illustrate its importance and present solutions, e.g., the Quantities, Units, Dimensions, and Types ontology

(QUDT). International scientific groups work on general solutions to the representation of physical units, such as the task group Digital Representation of Units of Measure (DRUM) from the Committee on Data of the International Science Council (CODATA).

### 3.4 Persistent identifiers

One problem that complicates the application of external databases is a non-uniform use of labels and identities (IDs) that generally can cause confusion and misunderstandings. Approaches for a harmonisation of IDs exist in different fields. For instance, the international generic sample number IGSN (Klump et al, 2021; IGSN; SESAR) provides persistent identifiers for materials and samples.

An aspect that is relevant for all types of studies involving measurements is the identification of instruments used for the studies. For a comprehensive documentation of a study, the instruments and their current states, i.e., version of software, date of last calibration, etc. need to be captured. A recent approach to document all necessary details about a measuring device is presented by the Research Data Alliance Working Group Persistent Identification of Instruments (PIDINST). Stocker et al. (2020) discuss the metadata schema that has been developed based on needs of the Earth science community. However, it is flexible enough to include all types of measuring devices, not limited to a specific discipline. A description of the PIDINST metadata schema is published and updated by working group members (Krahl et al., 2021).

The Open Researcher and Contributor ID (ORCID; Haak et al., 2012) allows indisputable identification of researchers, even in case of changing the name or the affiliation. For research institutes, an analogous identifier is provided with the Research Organization Registry ID (ROR). The funding of a work is not only identifiable through internal grant IDs from the according agencies, but also through persistent identifiers provided by, for instance, Crossref (Hendricks et al., 2020).

The examples mentioned above demonstrate that sophisticated solutions to different problems related with laboratory databases exist. Integrating these solutions into a geoscientific laboratory database model open to all disciplines in this field is a challenge we face in this study. In the next section, we describe our approach in more detail.

### 4 A conceptual complex database model for Geophysics

Our database model is set up as a modular system right from the start. One main goal was to implement a comprehensive set of metadata, allowing usability of datasets across disciplines. Metadata can be described as "information about the data" (Volk et al., 2014). We classify metadata into two groups: general and specific metadata. The group with general metadata comprises elements containing distinct parts of information related with different measurements and data processing procedures. Metadata about the investigated sample, the device used for the measurement, and the algorithms used for processing of the data, respectively, serve as exemplary elements in this category. In contrast, the elements considered as specific metadata, like information on the configuration of the measurement and details about the processing procedure, are unique. They arise for each measurement and data evaluation, respectively. In the following paragraphs, we take a closer look at the different elements in both groups of metadata and describe the links between these elements. A detailed list of the information contained in each element is given in the appendix (Table A1).

## 4.1 Classification of metadata

The elements from both groups, general and specific metadata, are linked with each other and form a network of metadata
referred to as data map (Fig. 1). The data map represents a simplified approach. For the sake of clarity, we do not map this
metadata network to existing international standards (e.g., ISO 19156). Nevertheless, these standards need to be considered
when a distinct database model is developed. According to the data map, two elements from the group of specific metadata
prove to be in a key position as they show the highest number of connections to other elements: measurement and processing
metadata.

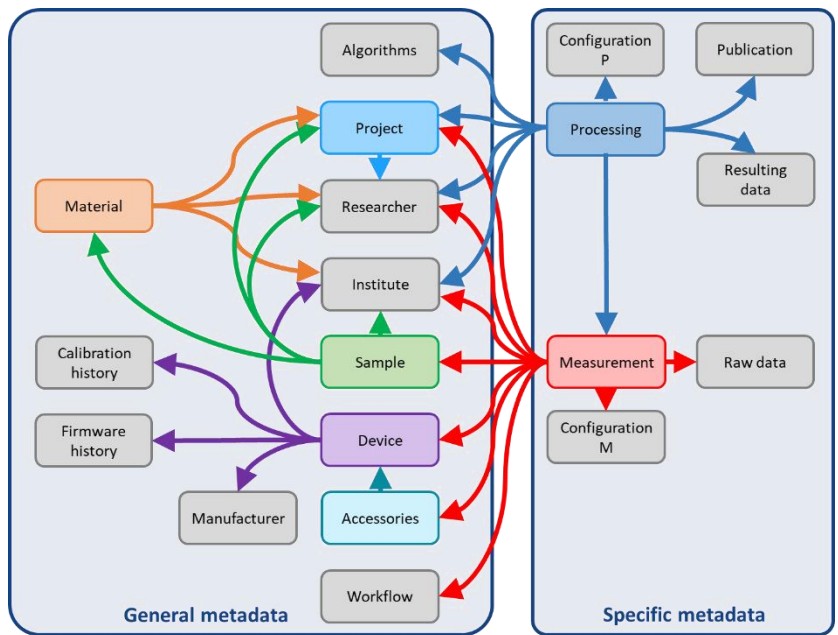

**Figure 1: Data map showing the separation of metadata into general metadata (left) applicable to several measurements and specific metadata (right) referring to a distinct measurement and the appropriate processing of the measured data. The arrows connecting the items indicate links between different sets of metadata. The grey boxes indicate that there are no further links from these items to others.**

### 4.1.1 Measurement metadata

The set of measurement metadata is the element with the highest number of links within the data map. Details on every
measurement (i.e., the name of the method, date of the measurement, and a link to the raw data) are stored in this element
under a unique ID. Additional links provide access to other essential elements, e.g., with details on the sample and the
measuring device (Fig. 2). The configuration of the measurement is described in a separate element that contains the set of
parameters necessary to replicate the measurement (Fig. 3). Despite the benefits of the UCUM system, using common
geophysical units here will improve acceptance of the model by the community. However, transformation into the UCUM

system is necessary when output in alternative units is required. As the number and type of relevant parameters vary depending on the applied method, we store the parameters in a table (used here as synonym for a database element) similar to an approach presented by Horsburgh et al. (2008) that allows assigning an undefined number of entries to each measurement. It is worth

205 noticing that for creating a well-planned relational database scheme, UML diagrams are required to map all dependencies between the different metadata. Specialised tools exist for this purpose. Figures 2 and 3 are intended to highlight the fundamental linkages and cross-relationships in between different types of data.

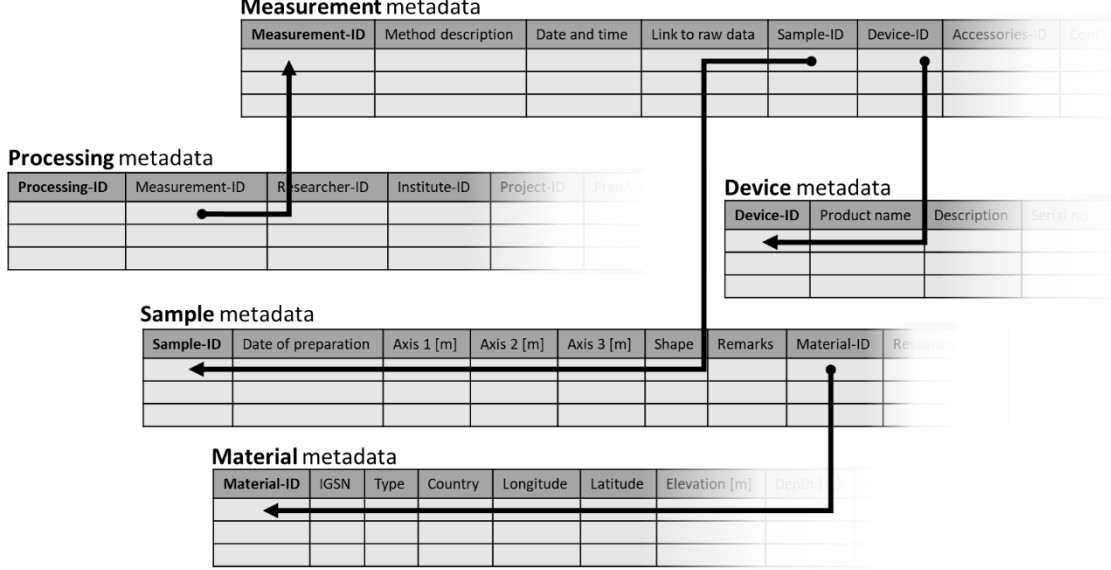

**Figure 2: Sketch of the tables containing different metadata on the measurement and the processing procedure, respectively. Variable numbers of parameters, depending on the applied method, can be stored in an arbitrary number of tables. Sorting of the**
210 **entries is enabled through the unique IDs in the first column of each table.**

## Configuration M metadata

| ConfigM-ID | Measurement-ID | Parameter Name | Parameter Value | Parameter Unit |
|---|---|---|---|---|
| 1 | 1 | Min. Frequency | 0.001 | Hz |
| 2 | 1 | Max. Frequency | 10000 | Hz |
| 3 | 2 | Magnetic Field | 400 | A/m |
| 4 | 1 | Max. Voltage | 10 | V |
| 5 | 2 | Min. Frequency | 505 | Hz |

**Figure 3: Sketch of the metadata element about the configuration of measurements. The first column ("ConfigM-ID") contains a unique ID for each entry. The second column indicates the measurement associated with the parameter details contained in the**
215 **residual columns. Exemplarily, we entered typical metadata related with two different methods, i.e., spectral induced polarisation (1) and magnetic susceptibility (2), respectively.**

The preparation of samples for geoscientific laboratory measurements marks an important step within the whole measurement workflow. As the description of laboratory procedures in geophysics is usually not standardized, clarity and machine-readability of this part of information is limited. We define a numerical code, where distinct numbers substitute each element of a sentence describing a step of the sample preparation (Fig. 4). Combining these numbers allows the construction of sentences of a defined length comprising subject, verb, preposition, object, and an expression for the duration, number of repetitions, and frequency, respectively. Thus, a detailed description of the workflow of sample preparation is made accessible to machines. Finally, additional information concerning the measurement, i.e., details on the related project, the responsible researcher, and the institute, are made accessible through links within the set of measurement metadata. With this, a high flexibility for complex workflows in research laboratories is achieved. Furthermore, standard workflows, e.g., defined by standard operating procedures (SOP) can be predefined and implemented within the final database. Editing and extending these workflows is possible at any time so that new laboratory methods and procedures do not compromise the database structure at all.

| Verb | | Subject | | Preposition | | Object | | Duration/Frequency | |
|---|---|---|---|---|---|---|---|---|---|
| 0 | void | 0 | void | 0 | void | 0 | void | 0 | void |
| 1 | dry | 1 | sample | 1 | in | 1 | sample | 1 | for 1 h |
| 2 | weight | 2 | tank | 2 | on | 2 | tank | 2 | for 6 h |
| 3 | saturate | 3 | measurement cell | 3 | between | 3 | vacuum | 3 | for 12 h |
| 4 | put | 4 | hose clamp | 4 | under | 4 | adapter | 4 | for 24 h |
| 5 | fasten | 5 | fluid conductivity | 5 | at | 5 | sleeve | 5 | for 1 week |
| 6 | insert | 6 | pH of fluid | 6 | of | 6 | fluid | 6 | one time |
| 7 | jam | 7 | column | 7 | with | 7 | climatic chamber | 7 | three times |

**Figure 4: Example for a numerical code to describe the workflow of sample preparation. For instance, the instruction "dry sample in vacuum for 24h" can be expressed as "1.1.1.3.4". If one element of the sentence is not applicable, it is expressed by "0" representing void space.**

### 4.1.2 Processing metadata

In Fig. 1, the set with the second largest number of connections to other elements is the processing metadata set. This element is also part of the group of specific metadata and provides all information related to the processing of the measured data. Beside links to the processed data and the responsible researcher, also details on the configuration of the processing procedure are stored in this element in the way illustrated in Fig. 2. Additionally, information on further use of the processing results, as the number of own publications containing the data, the limit of publications until the data will be published, and the latest date of data publication for this data are part of this set of metadata.

The algorithms used for processing and evaluating the measured data are essential parts of a laboratory study. To allow reproducibility of the results, these algorithms must be accessible to the scientific community. In case of algorithms published

under an open source license, this is easily feasible by providing links to the source code of the algorithms in the processing

metadata. If proprietary software is applied, a description of the software configuration and the underlying principles, including

appropriate references, is necessary to allow replication of the results. The data resulting from the processing procedure and

the publications produced from these results are inherently associated with the processing metadata. Therefore, a link to the

file containing the resulting data and a list of the according publications are part of the processing metadata.

### 4.1.3 Other metadata elements

Information on the measuring device and potential accessories, e.g., measuring cells, is registered in the respective metadata

sets. Beside the name of the device and the serial number, also the dates of initial setup and decommission (if applicable) are

registered in the device metadata set. Details on the manufacturer, e.g., contact information, are stored in a separate table linked

to the device metadata set. A variety of accessories may exist for each device. Therefore, we keep the information on the

accessories of measuring devices in a separate table (Fig. 5). The approach that we already used for storing the configuration

metadata provides the flexibility that is needed for handling the parameters related with accessories.

**Accessories** metadata

| AccessoryList-ID | Accessory-ID | Device-ID | Method | Parameter Name | Parameter Value | Parameter Unit |
|---|---|---|---|---|---|---|
| 1 | 1 | 1 | SIP | Length | 50 | mm |
| 2 | 1 | 1 | SIP | Inner Diameter | 20 | mm |
| 3 | 1 | 1 | SIP | Electrode Dist. | 30 | mm |
| 4 | 2 | 1 | SIP | Length | 55 | mm |
| 5 | 2 | 1 | SIP | Inner Diameter | 25 | mm |

**Figure 5: Sketch of the metadata element on the accessories of measuring devices. The first column ("AccessoryList-ID") provides a unique ID for each entry in the table. An arbitrary number of parameters can be registered for each accessory (column 2) belonging to a certain device (column 3). Exemplarily, we filled the table with notional data on two measuring cells for the method of spectral**
**induced polarisation (SIP).**

Whenever calibrations of the measuring instruments and updates of the according firmware, respectively, are needed, the date

of the calibration and details on the update (Fig. 6) must be archived to be able to reproduce a measurement. We keep this

information in separate lists that are accessible through the device metadata. In case of a faulty firmware update or a wrong

calibration of the instrument, affected measurements can be identified easily.


**Firmware history** metadata

| FirmwareUpdate-ID | Device-ID | Firmware version | Date of installation |
|---|---|---|---|
| 1 | 1 | 1.1.0 | 11.06.2019 |
| 2 | 2 | 1.0.0 | 12.03.2020 |
| 3 | 1 | 1.2.0 | 16.09.2022 |
| 4 | 1 | 1.2.1 | 20.09.2022 |
| 5 | 2 | 2.0.0 | 14.03.2023 |

**Figure 6: Sketch of the metadata element on the firmware update history. The first column ("FirmwareUpdate-ID") contains a unique ID for each entry. The second column indicates the device that received a firmware update. The current version of the firmware is described by its major, minor, and patch level, respectively, as demonstrated with notional entries in the table.**


Information on the investigated sample, i.e., its dimensions, shape, and the date of extraction are stored in the sample metadata. Details on the material of the sample are accessible through a link to an extra table, where the type of the material is specified together with petrographic and stratigraphic descriptions according to international stratigraphic classification standards (Cohen et al., 2013; Bär et al., 2020).

Information about the project that is related with the measured and processed data, respectively, can be found in a separate table. The table contains the title and a short description of the project, the name of the funding agency as well as the project ID given from the funding agency. Starting and ending dates of the funding period and the ID of the principal investigator in the researcher metadata set complete the information contained in this table.

### 4.2 Conversions between community and database language

To integrate different geoscientific disciplines in the laboratory database, a variety of common discipline-specific terms, parameters, and units must be considered from the start. Depending on the discipline, similar physical properties may be described by different parameters and units. To avoid misinterpretations, the content stored in the database has to be clearly defined. In this context, a distinction has to be made between data and metadata. As the data must be stored without any modifications, no transformation of parameters and units, respectively, can be performed. The data will be kept as provided

by the person in charge. Instead, metadata are intended to be accessible and searchable for every user. Therefore, metadata first need to be transferred from discipline-specific terms provided by the user to a harmonised set of parameters and units stored in the database. Schemas provided by international organisations (e.g., DataCite) and based on international standards (e.g., ISO 19115) should build the foundation of the metadata harmonisation. In case of a query, the metadata must then be transferred into discipline-specific parameters and units familiar to the user. Concerning the terms stored in the database,

discipline-specific thesauri can be used to perform the transfer from the content of the database to the discipline-specific expressions familiar to the respective users and vice versa. Morrill et al. (2021) present a thesaurus based on the Simple Knowledge Organisation System (SKOS, World Wide Web Consortium, 2009). This approach is not limited to a direct translation between two expressions as it allows the definition of hierarchical relations and the discrimination between

preferred and alternative expressions. As each thesaurus is defined specifically for a distinct discipline, the set of thesauri can be easily extended when a new discipline with its corresponding vocabulary is added to the database. Internationally accepted vocabularies that follow the FAIR data principles can be found for different disciplines in collections like Research Vocabularies Australia. The integration of existing vocabularies should be preferred instead of using individual word lists as it complies with the FAIR data principles. However, the selection of a suitable vocabulary must be done when a distinct geoscientific laboratory database is created. The implementation of already existing and established vocabularies is imperative when a specific community database is made available for other user communities. Configurability and extensibility of the thesauri provide the flexibility that is necessary in the context of a laboratory database open to all geoscientific disciplines.

Besides the terms and parameters used in the database, the units of physical quantities can vary between different geoscientific disciplines. The UCUM system (Schadow et al., 1999) is applied to harmonise the units stored in the database, facilitate machine-readability for automatic access following to the FAIR data principles, and simplify the transfer into discipline-specific units when queried by a user.

## 4.3 Reusability of data

Especially in the context of reusability, two aspects of a database model for research data become important: legal aspects of data sharing and the integrity and security of research data. Legal aspects not only cover copyright and licensing of the data, but also include questions on using open or proprietary data formats for storing and providing data. International standards (e.g., ISO 19153) provide information on the management of digital rights. However, this issue is too complex for an adequate consideration in this study on an initial conceptual model for geophysical laboratory data. We exemplarily refer to Carroll (2015) and Labastida and Margoni (2020) for further information. The issue of data security and integrity refers to mechanisms that prevent subsequent modification of the data once stored in the database. Although this aspect is vital for the reusability of research data, it is also beyond the scope of this paper. Nevertheless, both issues should be thoroughly addressed when designing a distinct database model.

## 5 Conclusions

Sharing and reuse of research data resulting from geoscientific laboratory measurements needs to be improved, to allow a sustainable handling of these highly valuable data. While excellent approaches exist for different geoscientific disciplines and individual methods, a general database open and applicable to laboratory data from all geoscientific disciplines is still missing. Such a database has to fulfil several requirements resulting from the intended interdisciplinarity, where extensibility of the database and conformability to discipline-specific particularities are the most prominent.

We present a conceptual model of a laboratory database intended for use in all geoscientific disciplines that is based on current approaches. The integration of recent concepts on workflow description, harmonisation of physical units, and thesauri provides the flexibility needed to handle the variety of terms, parameters, and units resulting from the wide field of application. Using

a relational database structure and a clear classification of metadata into different metadata sets allows extension of the database subsequently without modifying its structure. The database model was originally developed starting with geophysical laboratory methods. After the implementation, the database must prove its applicability to the variety of geoscientific data. Up to now, we did not consider legal aspects of data sharing in detail. The integration of this issue must be part of a future study. However, the due to its flexibility, the presented model will allow subsequent integration of legal aspects in the database, e.g., the consideration of rules for data publishing under certain conditions.

*Author contribution.* MH acquired the funding, proposed and managed the project, and edited the original and the revised draft. SN arranged the presented conceptual model, created the figures, and wrote the original and the revised draft.

*Competing interests.* The authors declare that they have no conflict of interest.

*Acknowledgements.* The authors would like to thank Veronika Grupp and the members of the "NFDI4Earth pilots" group for helpful discussions throughout the whole duration of the pilot project. The constructive comments of two anonymous reviewers improved the manuscript substantially.

*Financial support.* This work has been supported by DFG (Deutsche Forschungsgemeinschaft) within the NFDI4Earth initiative, grant no. 460036893.

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

**Appendix A**

**Table A1: List of all elements of the geoscientific laboratory database conceptual model.**

| Metadata element | Contents |
|---|---|
| Measurement | Measurement-ID, Sample-ID, Device-ID, Accessories-ID, Configuration-ID, Workflow-ID, Project-ID, Researcher-ID, Institute-ID, Method description, Date, Link to raw data |
| Processing | Processing-ID, Measurement-ID, Researcher-ID, Institute-ID, Project-ID, Preparation-Algorithm-ID, Evaluation-Algorithm ID, Publication-ID, ResultDataLink, Number of own publications, Publication limit, Date of latest publication, DOIs of publications |
| Sample | Sample-ID, Material-ID, Researcher-ID, Institute-ID, Project-ID, Date of preparation, Length (Axis 1), Width (Axis 2), Height (Axis 3), Shape, Remarks |
| Device | Device-ID, Institute-ID, Manufacturer-ID, Product name, Description, Serial number, Date of initial setup, Date of decommission |
| Material | Material-ID, IGSN, Researcher-ID, Institute-ID, Project-ID, Type, Country, Longitude, Latitude, Elevation, Depth, Date of extraction, Place of extraction, Petrographic description, Stratigraphic description |
| Project | Project-ID, Researcher-ID of PI, Title, Project description, Funding agency, Project-ID at funding agency, Funding period Start, Funding period End |
| Accessories | AccessoryList-ID, Accessory-ID, Device-ID, Method, Parameter Name, Parameter Value, Parameter Unit |
| Algorithms | Algorithm-ID, Programming language, Purpose, Link to algorithm |
| Configuration M | ConfigM-ID, Measurement-ID, Parameter Name, Parameter Value, Parameter Unit |
| Configuration P | ConfigP-ID, Processing-ID, Algorithm-ID, Parameter Name, Parameter Value, Parameter Unit |
| Institute | Institute-ID, Name of Institute, ROR/GRID, Department, Address, General phone number, General e-mail address, online presence |
| Researcher | Researcher-ID, Family name, First name, Middle name, ORCID ID |
| Workflow | Workflow-ID |
| Manufacturer | Manufacturer-ID, Name, Address, Contact info |
| Calibration history | Calibration-ID, Device-ID, Date of calibration, Remarks |
| Firmware history | FirmwareUpdate-ID, Device-ID, Firmware version, Date of installation |
| Publication | Publication-ID, Researcher-ID, Institute-ID, Date of publication, Type of publication, DOI of publication |