# Peer review of "Making geoscientific lab data FAIR: A conceptual model for a geophysical laboratory database"

_Geoscientific Instrumentation, Methods and Data Systems, 2023_

## Author Response (AR1)

**Detailed point-by-point response to all referee comments**

**on manuscript gi-2023-9**

**Comments by Referee No. 1**

**Referee's Comment (RC 1.1):** *This paper is well written and describes creating a conceptual model for a geophysical database. However, there already exists many papers in the literature and international ISO/OGC/W3C standards that set a higher-level abstract model that this paper needs to at least mention, if not consider and reframe their work into these standards. There are also existing papers on the use of Globally Unique Persistent Resolvable Identifiers and PIDs that are relevant to this paper. I feel that there needs to be a major revision of this paper, which puts the local context of the NFDI paper into this existing global framework, standards and vocabularies. I did not find all that many errors within the paper itself, more it was the lack of acknowledgement of what exists.*

*If the authors can frame their data model within these global contexts, then it will greatly accelerate FAIR compliance and machine-to-machine interaction of data stored within their proposed data system. I also predict it will lead to greater uptake of this paper and increase its scientific and technical significance and quality.*

**Authors' Response (AR):** We are grateful for the highly valuable comments and appreciate the additional sources recommended by Referee No. 1. After editing the manuscript, we now address international standards in the according sections of this paper. We now consider the existing persistent identifiers for instruments and funding suggested by referee no. 1.

**Authors' Changes (AC):** Relevant sections have been adapted and extended, respectively.

**RC 1.2:** *The abstract is clear (but needs referencing of the international frameworks), the writing and language is clear and precise, but there is just not enough referencing of existing work. Providing evidence of how this data model fits into existing higher-level ontologies and data models would increase my rankings of this paper substantially.*

**AR:** In the abstract, we now point to common standards and mention international working groups. References to distinct documents are placed in the appropriate sections in the text.

**AC:** [lines 10-14] **Global working groups as the Committee on Data of the International Science Council (CODATA) put effort in the development of tools improving research data handling. International standards (e.g., ISO 19156) and ontologies (e.g., UCUM) provide a general framework for certain aspects that are elemental for the development of database models. However, these abstract models need to be adapted to meet the requirements of the geoscientific community.**

**RC 1.3:** *Section 2.5, Line 96: I am very surprised that this section does not mention generic solutions (abstract models) that already exist for data models on this topic from ISO, OGC and W3C. These enable interdisciplinary integration of analytical and observational data and include ISO 19156:2023, Janowicz et al. (2018), Haller et al. (2018), Magagna et al. (2023). Papers 1-3 should be references in this paper.*

**AR:** We are thankful for pointing to the additional references. We now refer to the standard ISO 19156:2023 and the contributions by Janowicz et al. (2018). However, we were not able to find the publication by Magagna et al. (2023) in a peer-reviewed journal. Therefore, we found it difficult to cite this contribution.

**AC:** [lines 96-98] **International standards on metadata as ISO 19156 and sophisticated ontologies (e.g., Janowicz et al., 2018) enable linking to and exchange with other databases. By following such standards, a database model can be extended to fields beyond the original discipline.**

**RC 1.4:** *Section 3.3 Line 135: Units of Measure – the paper cites literature on Medicine. It does quote Hall and Kuster (2022) which mentions QUDT. However, this issue is widely known – I feel that there are other papers which should be quoted and maybe also the CODATA task group on the Digital Representation of Units of Measure (DRUM) which is working internationally with the International Science Council and affiliated Science Unions to get an agreed understanding and implementation of digital unit representation. This paper in particular could be cited: Hanisch et al. (2022).*

**AR:** We see that there are more recent contributions to this issue than we considered up to now. We include the paper by Hanisch et al. (2022) that provides an impressive example why a unified system for units of measure is urgently needed. We now reference QUDT.

**AC:** [lines 148-152] **However, the digital representation of physical units in databases is a problem that is relevant to all fields of science. Hanisch et al. (2022) illustrate its importance and present solutions, e.g., the Quantities, Units, Dimensions, and Types ontology (QUDT). International scientific groups work on general solutions to the representation of physical units, such as the task group Digital Representation of Units of Measure (DRUM) from the Committee on Data of the International Science Council (CODATA).**

**RC 1.5:** *Section 3.4 Line 141: Persistent Identifiers – To make activities in laboratories more machine readable and transparent and be able to trace contributions of researchers, funders and institutions this section should also include identifiers for funding (e.g., grant identifier of Crossref and RAiD). This should also include the recent use of Identifiers for Instruments proposal of the RDA Persistent Identifiers for Instruments Working Group and their outputs: Krahl et al. (2021), Stocker et al. (2020), McCafferty et al. (2023). This is a white paper, and hence not necessarily citable, but it also includes using PIDS for instrument calibration data.*

**AR:** We are thankful for your suggestion and include a reference to the mentioned identifier for funding by Crossref. We included reference to the proposal by the RDA Persistent Identifiers for Instruments working group and to Stocker et al. (2020). Thank you for pointing to the best practice white paper by McCafferty et al. (2023), which unfortunately cannot be cited.

**AC:** [lines 158-164] **An aspect that is relevant for all types of studies involving measurements is the identification of instruments used for the studies. For a comprehensive documentation of a study, the instruments and their current states, i.e., version of software, date of last calibration, etc. need to be captured. A recent approach to document all necessary details about a measuring device is presented by the Research Data Alliance Working Group Persistent Identification of Instruments (PIDINST). Stocker et al. (2020) discuss the metadata schema that has been developed based on needs of the Earth science community. However, it is flexible enough to include all types of measuring devices, not limited to a specific discipline. A description of the PIDINST metadata schema is published and updated by working group members (Krahl et al., 2021).**

[lines 167-168] **The funding of a work is not only identifiable through internal grant IDs from the according agencies, but also through persistent identifiers provided by, for instance, Crossref (Hendricks et al., 2020).**

**RC 1.6:** *Section 4.1, line 166: I would recommend that this figure be mapped to the ISO 19153 or SOSA models as these are widely accepted.*

**AR:** Figure 1 shows a map of the different groups of metadata that exist for geophysical laboratory measurements. The objective of this figure is to demonstrate that not a small set of metadata, but a complete network of information is required to describe the data resulting from a laboratory experiment. To highlight clearly the two sets of metadata that have a central position in this network, we chose a rather simple, more illustrative design for this map and did not use the abstract models defined by international standards.

**AC:** [added lines 185-187] **The data map represents a simplified approach. For the sake of clarity, we do not map this metadata network to existing international standards (e.g., ISO 19156). Nevertheless, these standards need to be considered when a distinct database model is developed.**

**RC 1.7:** *Section 4.2, Line 179: Conversion between community and database schemas. This paper should reference more of the exiting international metadata schemas that could be cited for this including ISO 19115 – Geographic metadata, DataCIte, Schema.org.*

*I would suggest that each term be looked at closely for an existing international standard. Many of these have internationally agreed and published vocabularies and definitions (e.g., Analytical Methods for Geochemistry and Cosmochemistry (https://vocabs.ardc.edu.au/viewById/650 ) -there are many other published relevant vocabularies in Research Vocabularies Australia (https://vocabs.ardc.edu.au/ ), the NERC Vocab server (https://www.bodc.ac.uk/resources/products/web_services/vocab/ ) and other vocabulary services.*

**AR:** We referenced DataCite and ISO 19115 providing the foundation of metadata harmonization. Below, we refer to the use of vocabularies

**AC:** [lines 287-288] **Schemas provided by international organisations (e.g., DataCite) and based on international standards (e.g., ISO 19115) should build the foundation of the metadata harmonisation.**

**RC 1.8:** *The FAIR principles are cited in line 268 and given that the FAIR principle 12 of Wilkinson et al (2016) states that all metadata and data use vocabularies that are themselves FAIR compliant and both machine and human actionable. For line 263 I would recommend that the authors look into what community vocabularies are already published and reference these. Even if these published vocabs do not cover all terms required by this paper, best community practice is to contact the authors of existing vocabularies and see if they will not extend these.*

**AR:** We now refer to Research Vocabularies Australia that provides different discipline-specific vocabularies. However, our conceptual model does not represent a distinct model for a geoscientific database. It is intended to be a more general, conceptual approach, targeting for a community with less or even no knowledge about FAIR data management. Therefore, we did not ask authors of existing vocabularies to extend their work. This needs to be done when a distinct geophysical laboratory database is developed.

**AC:** [lines 294-300] As each thesaurus is defined specifically for a distinct discipline, the set of thesauri can be easily extended when a new discipline **with its corresponding vocabulary** is added to the database. **Internationally accepted vocabularies that follow the FAIR data principles can be found for different disciplines in collections like Research Vocabularies Australia. The integration of existing vocabularies should be preferred instead of using individual word lists as it complies with the FAIR data principles. However, the selection of a suitable vocabulary must be done when a**

**distinct geoscientific laboratory database is created. The implementation of already existing and established vocabularies is imperative when a specific community database is made available for other user communities.**

**Comments by Referee No. 2**

**RC 2.1:** *I would expect a figure showing an actual relational database schema, possibly using Unified Modeling Language (UML), providing a comprehensive overview of the proposed database. Such a diagram would clarify the relationships between different components, such as the range of numbers of instances of components allowed in the relationship (e.g., 1 device can have N measurements). This diagram is essential for anyone looking to implement the proposed database. With such a figure, Figure 2, in particular, and potentially Figure 3 could be removed.*

**AR:** We can follow the referee´s comment on this. That being said, aim of this work, based upon an NFDI4Earth pilot project was the development of a conceptual model for a (national) community (geoscience in general, geophysics in particular), which is not really aware of state of the art FAIR data management and according database models. The authors of this manuscript are approaching this topic from the perspective of "users", not of database modelers / creators, in order to be as close as possible to our main group of stakeholders. Hence, we would like to keep these figures as they are, since they have proven to be "understood" by people, who would have to enter data there. We totally agree on the point that – given the opportunity to build such a database – a well-planned relational database scheme using UML tools (e.g., such as miro) is the way to go for.

**AC:** [lines 204-207] **It is worth noticing that for creating a well-planned relational database scheme, UML diagrams are required to map all dependencies between the different metadata. Specialised tools exist for this purpose. Figures 2 and 3 are intended to highlight the fundamental linkages and cross-relationships in between different types of data.**

**RC 2.2:** *In the example Figure 3, the units are in the International System of Units (SI), whereas the UCUM system is said to be applied (Line 267). This inconsistency should be addressed.*

**AR:** We addressed the apparent inconsistency and explained why we decided to use SI instead of UCUM in the text referring to Figure 3.

**AC:** [lines 200-202] **Despite the benefits of the UCUM system, using common geophysical units here will improve acceptance of the model by the community. However, transformation into the UCUM system is necessary when output in alternative units is required.**

**RC 2.3:** *In your introduction, you mentioned that extensibility and flexibility were key attributes of your approach. I wonder to what extent this applies. There is often a trade-off between flexibility and having a well-defined data structure for sharing data. Could you comment on how you balance these aspects in your design?*

**AR:** In a geoscientific laboratory database, a variety of discipline-specific methods, parameters, and units must be combined. Additionally, the database must be extendable when new methods, instruments, and experimental workflows, respectively, need to be integrated. Using a relational database is one essential aspect that provides the required extensibility without missing a well-defined data structure. For the acceptance of a database within a community that comprises different geoscientific disciplines, flexibility is another crucial aspect. Discipline-specific thesauri and ontologies like UCUM and QUDT, respectively, can help to provide an interface of the database that can be adjusted for each discipline without losing the obligatory data structure. Please see the Conclusions section.

**AC:** No changes.

**RC 2.4:** *When it comes to implementation, enabling global data sharing necessitates not only the use of common data formats but also the capability to read and reinterpret data, even over an extended period. Have you taken into account specific types of data formats or provided guidelines for their selection? I believe it would be valuable to mention these considerations.*

**AR:** Until now, we did not consider specific types of data formats. However, open formats should be preferred as they guarantee long-term access and readability. We now mention this aspect in the newly inserted section 4.3 on data security, integrity and legal aspects of data sharing.

**RC 2.5:** *Concerning the reusability of the data, I couldn't find any information regarding licensing or reusability conditions. Where are the metadata that provide the necessary information for making the data reusable, including origin and reuse conditions?*

**AR:** Legal aspects of data sharing are very important for reuse of data and must be considered when a database model is practically used. However, this issue is very complex and it is beyond the scope of our paper. Therefore, we decided to exclude it and to refer to the international standards and appropriate papers where legal aspects of data sharing are considered in more detail. Nevertheless, we acknowledge the importance of providing clear information on the origin of the data and reuse conditions. However, to emphasize the importance of this issue, we insert the new section 4.3.

**RC 2.6:** *In your discussions, did you consider data security topics, such as mechanisms for ensuring the security and integrity of the data, particularly in the context of tamper-proof data?*

**AR:** Like legal aspects of data sharing, also data security and integrity are important topics concerning a database model. We now refer to this issue in the new section 4.3

**AC:** [applies to RC 2.4, 2.5, and 2.6; lines 304-315]

**4.3 Reusability of data**

**Especially in the context of reusability, two aspects of a database model for research data become important: legal aspects of data sharing and the integrity and security of research data. Legal aspects not only cover copyright and licensing of the data, but also include questions on using open or proprietary data formats for storing and providing data. International standards (e.g., ISO 19153) provide information on the management of digital rights. However, this issue is too complex for an adequate consideration in this study on an initial conceptual model for geophysical laboratory data. We exemplarily refer to Carroll (2015) and Labastida and Margoni (2020) for further information. The issue of data security and integrity refers to mechanisms that prevent subsequent modification of the data once stored in the database. Although this aspect is vital for the reusability of research data, it is also beyond the scope of this paper. Nevertheless, both issues should be thoroughly addressed when designing a distinct database model.**